# Variation in the Promoter Region of the *MC4R* Gene Elucidates the Association of Body Measurement Traits in Hu Sheep

**DOI:** 10.3390/ijms20020240

**Published:** 2019-01-09

**Authors:** Girmay Shishay, Guiqiong Liu, Xunping Jiang, Yun Yu, Wassie Teketay, Dandan Du, Huang Jing, Chenghui Liu

**Affiliations:** 1Laboratory of Small Ruminant Genetics, Breeding and Reproduction, Huazhong Agricultural University, Wuhan 430070, China; shishay@webmail.hzau.edu.cn (G.S.); yunyu@webmail.hzau.edu.cn (Y.Y.); 2Key Laboratory of Agricultural Animal Genetics, Breeding and Reproduction of the Ministry of Education, Huazhong Agricultural University, Wuhan 430070, China; teketay@webmail.hzau.edu.cn (W.T.); dudandan@webmail.hzau.edu.cn (D.D.); huangjinghss@webmail.hzau.edu.cn (H.J.); liu890621@webmail.hzau.edu.cn (C.L.)

**Keywords:** body measurement traits, cis-elements, haplotype, *MC4R*, promoter, SNPs, transcription factor binding sites

## Abstract

The *melanocortin 4 receptor* (*MC4R*) gene is expressed in the appetite-regulating areas of the brain and is engaged in the leptin signaling pathway. Although previous studies have identified variants in the coding region of the sheep *MC4R* gene showing significant associations with birth weight, weaning weight, and backfat thickness, no such associations have been reported for the promoter region. Besides, the essential promoter region of the sheep MC4R has not been delineated. In this study, to better understand the transcriptional regulation of *MC4R* and to elucidate the association between regulatory variants and haplotypes with body measurement traits in sheep, we cloned and characterized the *MC4R* promoter. We found that the minimal promoter of the gene is located within the region −1207/−880 bp upstream of the first exon. Real-time quantitative PCR (RT-qPCR) data revealed the mRNA expression of the *MC4R* gene had a significant difference between sex and age. In the association analysis, eight single nucleotide polymorphisms (SNPs) had a significant association with one or more traits (*p* < 0.05); of these, two SNPs were novel. Notably, individuals with haplotype H1H2 (CT-GA-GT-GA-GT-GA-GA-CG) were heavier in body weight than other haplotypes. Altogether, variations in the *MC4R* gene promoter, most notably haplotype H1H2, may greatly benefit marker-assisted selection in sheep.

## 1. Introduction

Genetic markers are hallmarks of molecular variation, which are highly associated with the economically important traits of farm animals. The application of these markers via marker-assisted selection has the potential to alter the rates of genetic improvements [1]. Nowadays, the advancement of growth-related traits in livestock is at the most standing position in the meat industry because of the ever-increasing demand for meat and meat products around the globe. However, improvement of the favorable traits based on classical animal breeding programs would be difficult because most of the production traits are complex and controlled by multiple genes. Hence, understanding the genetic basis of these traits is essential to efficiently lead genetic improvement based on the candidate gene approach [2].

In the last few years, it has become increasingly evident that several genes have been reported to take part in the neural signaling pathway of energy homeostasis [3,4]. Among these, the *melanocortin 4 receptor* (*MC4R*) gene encoding a G-protein coupled receptor expressed in the appetite-regulating areas of the brain has been the object of ample investigations [5].

The *MC4R* gene signaling induces expression of a group of specific genes to exert catabolic effects through decreasing voluntary feed intake and increasing energy expenditure [6]. In mice, targeted disruption of the *MC4R* gene has resulted in an obesity syndrome characterized by hyperphagia, hyperinsulinemia, hyperglycemia, and increasing growth without affecting the reproductive axes [7]. Besides, the heterozygotes mutations on the *MC4R* locus in human and mice studies exhibit a phenotype intermediate between that of wild-type and the homozygous counterparts [8,9]. The involvement of the *MC4R* gene in the mediation of the effects of melanocortin on appetite regulation in humans and mice is well established [10]. Moreover, variants in the *MC4R* gene have been reported to be significantly associated with various traits that have paramount importance in livestock production such as in cattle [3,11], pig [12,13,14], sheep [15,16], and goat [17]. 

Nevertheless, the majority of the studies carried out in the *MC4R* gene were limited to the coding region variants and splicing sites of the gene that affects mRNA splicing, and protein function. However, the promoter region variants may also directly influence the transcriptional regulation of a gene through changing the binding sites of specific transcription factors, which are unswervingly involved in the initiation of gene transcription [18,19]. Thus, genetic variants in the promoter region are incredibly relevant to economically valuable traits in livestock [20,21]. Despite that a considerable number of polymorphisms in the coding region of the sheep *MC4R* gene revealed significant associations with birth weight, weaning weight, and backfat thickness, no such associations have been reported for the promoter region. Moreover, the sheep *MC4R* gene promoter has not yet been delineated. 

Thus, from this point of view, this study was carried out in Hu sheep to characterize the essential promoter region of the *MC4R* gene and identified its regulatory variants associated with various body measurement and fat deposition traits.

## 2. Results

### 2.1. Single Nucleotide Polymorphism (SNP) Identification

The sheep *MC4R* gene is located on chromosome 23, and it encodes 2 exons and 332 amino acids. In this study, regulatory variants in the 5′ region of the sheep *MC4R* gene were screened by DNA pool sequencing, and eight SNPs were identified (−1131C>T; −1038G>A; −1036G>T; −1026G>A; −943G>T; −287G>A; −206G>A; and −103C>G) (Appendix A). Among these, two SNPs (–1038G>A and −943G>T) were novel and have been deposited in the European Variation Archive, with accession numbers PRJEB29384 and ERZ778075, respectively. 

Here, in our study, the genotype, minor allele frequencies (MAF), and diversity parameters of the sheep *MC4R* are presented in Table 1. There were three genotypes for each locus, except −206G>A, for which the AA genotype was not observed. The analysis of identified polymorphic sites also showed that the polymorphism at nucleotide position −287 (−287G>A), −206 (−206G>A), and −103 (−103G>C) relative to the first nucleotide of the start codon was detected within the range of CpG Island. All eight variants were individually genotyped for the association analysis. The Chi-square tests in this population showed that most of the SNPs (five out of eight SNPs) were satisfying the Hardy–Weinberg equilibrium (HWE) (*p* > 0.05). However, the remaining three SNPs significantly deviated from the HWE (*p* < 0.05), indicating that the distribution of alleles varied from one generation to another [22]. All the polymorphisms detected in the 2.00 kb 5′ flanking region of the sheep *MC4R* and their frequencies (genotype and MAF) are shown in Table 1. 

### 2.2. SNP Variation in Potential Cis-Regulatory Elements of the MC4R Gene

The ConSite database (http://consite.genereg.net/cgi-bin/consite) [23] was used to identify the potential cis-regulatory elements in the 5′ flanking region of the sheep *MC4R* gene; we detected 574 potential cis-regulatory elements in the 5′ flanking region (2.00 kb) (data not shown). Therefore, those recognition sequences containing one or more SNPs were evaluated. Accordingly, a total of 19 known regulatory motifs changed as a result of the SNPs detected in the sheep *MC4R* promoter region. The regulatory motifs (transcription factors) were scored on both strands of the sheep MC4R sequence, of which 14 were in the plus and 5 in the minus strands. Detailed information on the SNPs and motifs are present in Appendix A.

### 2.3. Association Analyses

The association analyses between various body measurements (BMTs) and fat deposition traits with the eight variants in the regulatory region of the sheep *MC4R* are shown in Table 2. Of the eight SNPs, after Bonferroni correction for multiple testing, seven SNPs (−1131C>T, −1036G>T, −1026G>A, −943G>T, −287G>A, −206G>A, and −103C>G) remained significantly associated with rump length (RL) and rump width (RW) (*p* <0.01), five SNPs showed a highly significant association with body weight (BW) and body length (BL) (*p* <0.05), three SNPs with lip% (*p* <0.05), and other two SNPs (−1036G>T, and −206G>A) were strongly associated with backfat thickness (BFT) (*p* <0.05). All eight identified SNPs were significantly associated with one or more body measurement trait (*p* < 0.05). In contrast, no SNPs were significantly associated with rump height (RH) and loin eye area (LEA) in this population, so we excluded both traits from further analysis (Table 2). 

The least-squares mean (LSM) of the regulatory variants showed that the variant -1038G>A was significantly associated with body weight and lip% traits (*p* < 0.05). The Duncan’s mean comparing analysis in analysis of variance (ANOVA) revealed that the Hu sheep with the heterozygous genotype, GT, at the −943G>T locus had a significantly higher BW than those with the homozygous GG and TT genotypes. On the contrary, the sheep with the TT genotype possessed the highest RL, RW, and lip% (*p* < 0.05). At the variant −206G>A, the heterozygous (GA) carrying animals had higher average body weight (25.89 ± 0.45 kg), RL (15.41 ± 0.14 cm), and RW (14.43 ± 0.13 cm), as well as higher BFT (0.73 ± 0.02 cm) than the wildtype genotype (GG). Similarly, at the variant (−103C>G), individuals with GG genotype possessed higher body weight (26.65 ± 1.25 kg) and higher lip% (7.17± 0.81) accumulation in the wool of the sheep than the other genotypes (CC and CG) sheep. 

### 2.4. Linkage disequilibrium and Haplotype Analysis of the MC4R Gene

Haplotype block and linkage disequilibrium (LD) structures were generated from the eight SNPs genotyped in the *MC4R* gene from Hu sheep (Figure 1). The LD plot presents the seven SNPs located in the promoter region of the sheep *MC4R* gene. It possesses two distinct LD blocks composed of the first four SNPs and the last three SNPs, thus the seven SNPs comprise a complete LD block. The darker shading indicates higher linkage disequilibrium and the number within rhombus is *D’* value. The set of haplotypes for the variations and their population frequency are indicated in a light gray color.

As shown in Appendix A, the degrees of linkage disequilibrium information between the eight *MC4R* SNPs revealed that the *D’* values ranged from 0.542 to 0.9996, and the *r*^2^ values ranged from 0.4978 to 0.8889. The higher degrees of LD existed between SNP1–SNP4, SNP4–SNP8, SNP6–SNP8, SNP4–SNP8, SNP6–SNP8, SNP4–SNP6, and SNP1–SNP8, as indicated by the higher *r*^2^ values (*r*^2^ = 0.8889, 0.8839, 0.8783, 0.8839, 0.8783, 0.8698, and 0.818, respectively). In this study, all the values *r*^2^ > 0.33, suggesting that the LD is sufficiently strong for use in gene mapping [24].

Furthermore, the haplotype analysis of the population was performed using the SNPstat online server (https://www.snpstats.net/analyzer.php) [25]. In this Hu sheep population, eleven haplotypes were identified. The frequencies of the haplotypes ranged from 0.0102 to 0.6321, and H1 had the highest frequency (0.6321) (Appendix A). Because we were interested in common genetic polymorphisms with frequency >0.05 [26], all haplotypes with frequencies <0.05 were excluded from our analysis. Therefore, three haplotypes, namely, H1 (CGGGGGGC), H2 (TATATAAG), and H3 (TATATAGG), and three diplotype were used in the correlation analysis (Table 3). The sheep with the H1H2 (CT-GA-GT-GA-GT-GA-GA-CG) diplotype had heavier body weight and backfat thickness (BFT) than those with the other haplotypes. Conversely, the individuals with the H3H3 (TT-AA-TT-AA-TT-AA-GG-GG) diplotype were preferred for their higher lipid percentage (LIP%) on their wool than the other littermates.

### 2.5. Sequence Analysis of the Promoter Region of MC4R Gene

In this study, the sheep *MC4R* gene showed several cis-regulatory motifs including for CCAAT boxes positioned at −495/−510 and −855/−870, TATA Box (TATAAA) consensus at position −427/−441, −837/−851, −1151/−156, −1582/−1587, and −1655/−1660 bp relative to the first nucleotide of the first exon of the sheep *MC4R* gene (Appendix A), and a CpG Island located within the region −60/−293 bp relative to the first nucleotide of the first exon (Figure 2). Furthermore, in the examined 5′UTR fragment (−2000 to +88) of the *MC4R* gene, we detected various putative cis-regulatory elements such as transcription factor binding sites for MyoD, GATA-1, GATA-2, C/EBP, HFH-2, HNF-3beta, FOXD3, cap, C/EBPbeta, CREB, CRE-BP1, Nkx, Nkx2-5, AP-1, SPI-B, and SPI-1. Notably, the upstream regulatory regions from −404 to −1980 and +104 to −1429 for sheep showed the highest number of predicted transcription factor binding sites (TFBSs) (Appendix A).

### 2.6. Differential Expression of the MC4R Gene across Age and Sex

To determine whether there is a difference in mRNA expression of *MC4R* between sex and age in different tissues, we analyzed the relative mRNA expression using real-time quantitative PCR (RT-qPCR). The results presented in Figure 3 demonstrated that in the lamb group, the relative mRNA expression of *MC4R* was significantly affected by sex (*p* < 0.05). In male lambs, remarkably higher mRNA expression was detected in the hypothalamus as compared with female lambs, whereas the opposite result was found in the kidney. In the heart, lung, liver, semimembranosus muscle, and longissimus muscle, sex had no significant effect on mRNA expression of the *MC4R* gene.

The data shown in Figure 3 indicated that in the adult group, sex had a significant effect on the hypothalamic, kidney, and liver mRNA expression of the *MC4R* gene (*p* < 0.05). The hypothalamic and liver mRNA expression of *MC4R* was substantially higher in male than in female littermates, whereas the opposite result was found in the kidney. In the heart, lung, semimembranosus muscle, and longissimus muscle, a significant difference in mRNA expression of the *MC4R* gene was not detected between sex groups.

### 2.7. The Proximal Minimal Promoter Region of the MC4R Gene

To measure the activity of potential cis-acting elements and to determine the minimum sequence required for *MC4R* activity, we generated a series of five reporter constructs with progressively larger deletions from the 5′ end of the promoter. The activities of these plasmids were evaluated upon transfection of the corresponding luciferase reporter plasmids into human embryonic kidney cells (HEK293) and mouse myoblast cells (C2C12), and the results of these analyses are shown in Figure 4. The luciferase activities of the 5′ flanking plasmids from pGL3-2000 to pGL3-369 were all higher than the negative control pGL3-basic. Notably, the activity of pGL3-2000 was eight to nine times higher as compared with the empty vector in the two cell lines, indicating that the functional promoter is in the −2000/+88 region of the *MC4R* gene. However, when the promoter was deleted to position −1489/+88, its activity decreased by 11.11% in HEK 293 cells and by 25% in C2C12 cells compared with the pGL3-2000/+88. Further, deletion of the promoter to −1207/88+ increased the promoter activity by 2.77-fold in C2C12 cells and by 4.5-fold in HEK 293 cells compared with the pGL3 −1489/+88. However, the further deletion to pGL3−880/+88 diminished the promoter activity by 38.88% in C2C12 cells and by 34% in HEK293 cells compared with the pGL3−1207/+88 in the two cell lines; this result demonstrated that the positive regulatory elements of the *MC4R* gene promoter are in the −1207/−880 region. Further deletion of the promoter to pGL3−369/+88 did not significantly change promoter activity. These results suggest that the minimal active promoter of *MC4R* gene is in the −1207/−880, which contains the consensus motifs for AP-1, SP-1, C/EBPalp, Hunchback (Zn-Finger, C2H2), HMG-IY (HMG), and HNF-3beta (Forkhead) (Figure 4). 

## 3. Discussion

Livestock body measurement traits mostly depend on the regulation of feeding behavior and fat metabolism [27]. For instance, *MC4R* signaling mediates the effects of leptin on food intake, which can regulate fatness and body weight traits [28]. These traits merely subjected to the precise control of gene expression at the transcription level [29]. Thus, genetic variants localized on these regulatory regions have practical implications for livestock breeding programs.

An important research finding was reported regarding the role of *MC4R* gene polymorphisms on body weight control and fatness traits in livestock, as well as on early obesity in humans. To date, more than 166 mutations were reported in human *MC4R* locus, mainly associated with growth and obesity [30]. However, most of the functional analyses of these variants have focused on the CDs region. Specifically, in sheep, the essential promoter region of the *MC4R* has not been delineated, and the functional analysis of genetic variants associated with birth weight, weaning weight, and backfat thickness at the locus has so far been constrained to coding region variants. In our current study, we characterized the essential promoter region of the *MC4R* gene and identified its regulatory variants associated with various body measurement and fat deposition traits.

In this study, we reported 8 SNPs and 11 haplotypes in the sheep *MC4R* promoter region, which altered 19 TFBSs; among these, three SNPs (−287G>A, −206G>A, and −103C>G) were located within the CpG Island region of the gene and two SNPs, SNP2 (−1038G>A) and SNP5 (−943G>T), were reported for the first time. 

The association analysis between the MC4R gene polymorphism with various body measurements and fat deposition traits was performed in 206 animals (Table 2). The overall results were concordant with previous studies in the coding region of the *MC4R* gene association reports [31,32,33,34]. The recent study in Hu sheep *MC4R* gene noted that similar to the polymorphisms in the coding region, variants in the regulatory regions are also exhibit a significant association with the body growth and meat quality traits of Hu sheep [35]. Likely, in this study, all eight identified SNPs show a significant association with one or more linear body measurement traits. Our data demonstrate that the genetic variants in the essential promoter region of the *MC4R* have a substantial effect in the *MC4R* gene activity. Mainly, the five SNPs revealed a statistically significant association with the most economically important traits (BW and BL) and the other two SNPs exhibited a strong significant association with BFT, which may indicate the relevance of regulatory variants in the *MC4R* locus as a gene marker in sheep body weight and meat quality enhancement programs.

Similarly, Duncan’s mean comparison analysis in ANOVA revealed that Hu sheep carrying the H1H2 (CT-GA-GT-GA-GT-GA-GA-CG) diplotype showed heavier BW and a higher backfat thickness than individuals with other haplotypes.

Conversely, the sheep with H3H3 (TT-AA-TT-AA-TT-AA-GG-GG) diplotype presented higher wool lipid accumulation than other haplotype combinations. These findings provide evidence to support that the *MC4R* gene along with its variants has a significant effect on body weight (BW), backfat thickness (BFT), and lipid percentage of wool in sheep. Our findings may greatly benefit the marker-assisted selection (MAS) programs in commercial sheep lines, to optimize desirable traits in sheep.

Besides, to characterize the minimal promoter region that can contribute to the transcriptional regulation of the *MC4R*, we cloned and characterized the potential promoter sequence of the gene using a series of 5′ promoter deletion analysis. Our results revealed the sheep *MC4R* promoter sequence contained one transcription start site (TSS), a CpG island, and the minimal functional promoter region closed to the TATA box consensuses. Moreover, the sheep *MC4R* gene promoter region unveiled numerous essential cis-regulatory elements shared with human and pig; similar promoter region sequences including the cAMP response element binding (CREB) protein, transcription sites activators, enhancers, GATA, and homeoboxes; and a response element for the neural zinc finger family of transcription factors (Appendix A) [36]. 

The *MC4R* gene belongs to the superfamily of G protein-coupled receptors (GPCR) an analogy to β-adrenergic receptors regulate intracellular cAMP concentrations by G protein-mediated adenylyl cyclase activation [37]. The effects of *MC4R* on gene expression have so far been attributed to cAMP-mediated PKA activation leading to subsequent phosphorylation of the transcription factor CREB and CRE-dependent transcription [38]. The CRE is commonly expressed in a variety of cell types. It has a well-documented role in cAMP, induces phosphorylation of CRE, and thereby activates the cAMP-responsive genes, to promote cell proliferation, differentiation, or modulation of various cell functions [39]. In this experiment, the CRE-like sequence (TTTGATGTAATC) and the CRE-BP1 (TTACATTA) were detected at two positions. In the first, CRE is positioned at −1756 to −1768 bp, and in the second, CRE-BP1 is present at −1281 to −1288 bp in the sheep promoter sequence. This sequence also contains a potential binding site (AttGGTCA) for monomeric nuclear receptors, such as thyroid receptor or steroidogenic factor 1 (SF1), activator protein 1 (AP-1), estrogen response element (ERE), myogenin/MyoD family (Myf), and myocyte enhancer factor 2 (Mef-2). This result is in agreement with previous findings by the authors of [40], who reported the important transcription factor binding site for the C/EBPβ, SRBP, SRF, factor1, MCM1, a band I factor, and DBP spanning in the 5′ flanking region of the sheep *MC4R* gene. These TFBSs are mainly involved in the regulation of several lipid metabolism pathways.

Notably, the sets of E-box motifs found in this promoter sequence essentially act as protein-binding sites, which are important for gene expression regulation in neurons, muscles, and other tissues [41]. Usually, these sequences bind with basic helix–loop–helix (bHLH) types of transcription factors. Once these sequences are recognized and bounded by the transcription factor, other enzymes can bind to the promoter and facilitate transcription from DNA to mRNA [42]. Hence, as the presence of such motifs enhances the transcriptional activity of the gene, a change in these motifs may also disrupt the transcriptional regulation of the gene. Such mechanisms may eventually lead to declining basal transcriptional activities. The previous study in the human *MC4R* transcriptional regulation [43] reported that the nescient helix–loop–helix 2 (Nhlh2) transcription factor directly influences the human and rodent body weight control pathways. Our finding is also consistent with previous reports in the promoter region variants of another genes [44,45,46], reported that the SNPs found in the promoter region revealed a significant association with the majority of economically important traits in livestock. These findings may support the evidence that the promoter variants altering the cis-acting elements could considerably affect the phenotype of an individual [47]. Therefore, as the same mutations in the human *MC4R* gene are well recognized as an infrequent cause of obesity [48,49], it is credible that SNPs in this proximal minimal promoter region of the sheep *MC4R* could also contribute to the body weight and fat deposition traits of the sheep. Collectively, our data suggested that the SNPs found in the promoter region, as well as their haplotypes, significantly affect the body measurement and fat deposition traits of sheep. Finally, our data demonstrated that identifying and characterizing the genetic variations in the *MC4R* promoter region in Hu sheep is preliminary and crucial for utilizing the genetic merits of Hu sheep in future genetic improvement programs. Therefore, this study may promote breeding of the ideal meat-type sheep in Chinese conditions using the best available genetic variants.

## 4. Materials and Methods 

### 4.1. Ethical Statement 

This study was carried out under the guidelines for the care and use of animals for scientific purposes set by the Ministry of Science and Technology, Beijing, China (No. 398, 2006). The protocol was approved on 03/07/2017 by the Institutional Animal Care and Use Ethics Committee of Huazhong Agricultural University, with the permit number for conducting animal experiments of (HZAUGO-2017-005).

### 4.2. Sampling and DNA Extraction

This study involved 206 Hu sheep kept at Huazhong Agricultural University (HZAU) Practice Teaching Center, Hubei fecund sheep breeding farm (Yichang, China). All animals were kept under semi-open sheds and maintained under the same system of feeding in the farm. The growth and bodyweight measurement traits, such as body weight (BW), withers height (WH), heart girth (HG), body length (BL), rump height (RH), rump length (RL), and rump width (RW), were measured using veterinary meter tape and sensitive weight ground balance. The backfat thickness (BFT) and loin eye area (LEA) were measured at between twelfth and thirteenth rib with Aquila veterinary ultrasound scanner [23]. The lipid percentage (LIP%) of the wool was also measured using the SOXHLET EXTRACTION METHOD EPA 3540C in HZAU, College of Animal Science and Technology Livestock Nutrition Laboratory. Blood samples for DNA extraction were collected, following published protocols [50].

### 4.3. Identification of Single Nucleotide Polymorphisms (SNPs)

PCR products from DNA of 206 sheep were used to identify polymorphisms within the 5′ flanking region of the *MC4R* gene. Using available reference sequences of sheep GenBank database (NCBI accession number: NM_001126370), primer pairs (Appendix A) were designed to amplify overlapping PCR products of the 2.00 kb region upstream of ATG.

DNA sequencing was performed by Quintarabio Company (Wuhan, China). Reference sequences and sequenced fragments were analyzed using version 12 DNASTAR™ Lasergene Genomics Suite Software, Madison, WI, USA. Finally, the nucleotide sequence of Hu sheep *MC4R* promoter partial sequence was submitted to GenBank under accession number MG970686.

### 4.4. RNA Extraction and Real-Time qPCR

To detect the relative mRNA expression of MC4R, we collected seven tissues (hypothalamus, kidney, liver, heart, lung, semimembranosus, and longissimus muscles) from Hu sheep from birth to six months (one, two, three, four, five, six months) of age (*n* = 42). Total RNA from each tissue was isolated using TRIzol reagent (Invitrogen, Carlsbad, CA, USA) according to the manufacturer’s protocol. Then, the RNA samples were used for complementary DNA synthesis using the reverse transcription kit Hifair TM II 1st Strand cDNA Synthesis Supermix kit with gDNA Eraser (YEASEN, Shanghai, China). Real-time quantitative PCR (RT-qPCR) was performed in an Applied Biosystems thermocycler (ABI7500, Thermo Fisher Scientific, Waltham, MA, USA) using the HieffTMqPCR SYBR Green Master Mix (YEASEN, Shanghai, China). The bovine β-actin gene was used as an endogenous control, using the primer sets (Appendix A). The relative expression of mRNA was calculated using the 2^−ΔΔ*C*t^ methods [51]. 

### 4.5. Sequence Characterization of the Potential Promoter Region of MC4R in Sheep

The promoter sequences prediction was carried out using Berkeley Drosophila Genome Project (http://www.fruitfly.org/seq_tools/promoter.html) [52], prediction of transcription start sites (TSS) with Promoter 2.0 Prediction Server (http://www.cbs.dtu.dk/services/Promoter/) [53], and the EMBOSS Cpgplot (https://www.ebi.ac.uk/Tools/seqstats/emboss_cpgplot/) [28] for the prediction of the CpG Island. Moreover, the transcription factor (TF) binding sites were analyzed with rVista 2.0 (http://rvista.dcode.org) [54,55] and CONREAL web server (http://conreal.niob.knaw.nl) [56]. 

### 4.6. Promoter Cloning and Generation of Luciferase Reporter Constructs

To clone the promoter region, we designed gene-specific primers to amplify a 2.00 kb genomic region upstream of the first nucleotide of the first exon of the sheep *MC4R* gene. The PCR product was isolated from agarose gel using a gel extraction kit (omega Bio-Tek, Inc., Norcross, GA, USA) and was cloned into pDrive (cloning vector). For the generation of the luciferase reporter construct, the (−2000 to +88) bp of the sheep *MC4R* promoter fragment was excised from the pDrive (cloning vector) by digestion with xho I and kpn I (Takara, Dalian, China), and ligated into the pGL3-basic vector digested with the same restriction enzymes. This plasmid was named pGL3-2000. Plasmids pGL3-1489, -1207, -880, and -369, which contained unidirectional deletions of the promoter, were generated by PCR using specific primers (Appendix A) with the sequence of the Xho I and kpn I restriction sites incorporated and pGL3-2000 as a template.

### 4.7. Cell Culture, Transfection, and Luciferase Assay

Human embryonic kidney cells (HEK293) and mouse myoblast cells (C2C12) were maintained in Dulbecco’s modified Eagle’s medium (DMEM; Invitrogen, Waltham, MA, USA) with 4500 mg/L glucose. The medium was supplemented with 10% fetal bovine serum (FBS) (PAA, Austria), 100 units/mL penicillin (final concentration), and 100 mg/mL streptomycin. Then, the cells were incubated at 37 °C with 100% humidity under 5% CO_2_ and passaged using standard cell culture techniques. Briefly, cells were seeded in six-well plates cultured at 37 °C with DMEM containing 10% FBS, but without antibiotics. When the confluence reached 80% to 90%, the cells were transfected with the plasmids described above using Lipofectamine 2000 transfection reagent (Bethesda, Maryland, USA), according to the manufacturer’s instructions. At 48 h after transfection, the cells were washed with PBS and total lysates prepared using passive lysis buffer (Promega Corp, Fitchburg, WI, USA). Luciferase activity was measured using the dual reporter assay system (Promega Corp) and EnSpire™ Multilabel Reader Valid for instruments with software version 3.0 Wallac Oy, Mustionkatu 6, FI-20750, Turku, Finland. The levels of firefly luciferase activity were normalized to Renilla luciferase activity and expressed as arbitrary units. Finally, the ratio of firefly luciferase light units to Renilla luciferase light units was analyzed, which included three independent experiments, following published protocols [57].

### 4.8. Statistical Analysis 

Allele and genotype frequencies and Hardy–Weinberg equilibrium (HWE) test were determined using Genalex 6 [58]. Haplotype analysis was performed using GEVALT (GEnotype Visualization and ALgorithmic Tool) [59]. The effect of haplotypes was analyzed using SNPstat web tool (https://www.snpstats.net/start.htm).

### 4.9. Association Analysis

The association between SNP marker genotypes of the *MC4R* gene and BMTs (BW, WH, HG, BL, HR, RL, RW, BFT, LEA, and LIP %) were analyzed using SPSS software (version 20.0, IBM Corp, Armonk, NY, USA) according to the following statistical linear model: (1)$$Y_{ijl}=\mu +G_{i}+S_{j}+A_{l}+E_{ijl}$$
where *Y_ijl_* is the observation of the BMTs and fat deposition traits measured on each of the ijlth animals, *μ* is the overall mean, *G_i_* is the genotype effect, *S_j_* is the fixed effect of sex, *A_l_* is the fixed effect of age, and *E_ijl_* is the random error. All data were described as least-square means ± standard error of means (LSM ± SEM). The values of the LSM were compared for significance using Duncan’s test. Differences were considered to be significant at *p* < 0.05.

### 4.10. Expression Analysis 

The relative expression level of *MC4R* was calculated using an Applied Biosystems thermocycler (ABI7500, Thermo Fisher Scientific, Waltham, MA, USA) by normalizing to the expression of β-actin with the 2^−ΔΔ*C*t^ method. The differences of mRNA expression level among age and sex were estimated by one-way ANOVA using Graph Pad Prism 5.0 software (San Diego, CA, USA). The results of the multiple comparisons were corrected by Bonferroni correction, and the differences were considered significant if *p* < 0.05.

## 5. Conclusions

Our study revealed that sheep *MC4R* is highly expressed in hypothalamus, kidney, and liver tissues and that multiple transcription factors mediate its expression. In this study, the proximal minimal promoter region was delineated, and 8 regulatory SNPs and 11 haplotypes in the *MC4R* gene of Hu sheep were identified; among these, the two regulatory variants (−1038G>A and −943G>T) were novel. Remarkably, all the identified SNPs were significantly associated with one or more linear body measurement traits (*p* < 0.05). Finally, we find the sheep with the H1H2 (CT-GA-GT-GA-GT-GA-GA-CG) diplotype had heavier body weight than those with the other haplotypes. Therefore, variants in the *MC4R* promoter region, most notably haplotype H1H2, may serve as a genetic marker to optimize breeding programs for body measurement traits in Hu sheep. In conclusion, this is the first report on the promoter region of the *MC4R* gene in sheep showing an association with economically important traits of sheep, which will deepen our understanding of how these cis-regulatory elements are involved in regulating the gene expression.

## Figures and Tables

**Figure 1 ijms-20-00240-f001:**
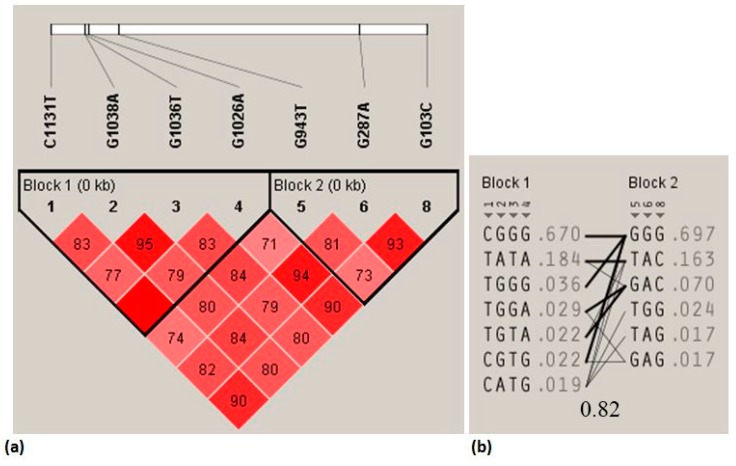
Linkage disequilibrium (LD) plot displays seven single nucleotide polymorphisms (SNPs) located in the promoter region of the sheep *melanocortin 4 receptor* (*MC4R*) gene. (**a**) There exist two different LD blocks composed of the first four SNPs and the last three SNPs, thus the seven SNPs comprise a complete LD block. The darker shading indicates higher linkage disequilibrium and the number within rhombus is *D’* value. (**b**) The sets of haplotypes for the variations and their population frequency are indicated in a light gray color.

**Figure 2 ijms-20-00240-f002:**
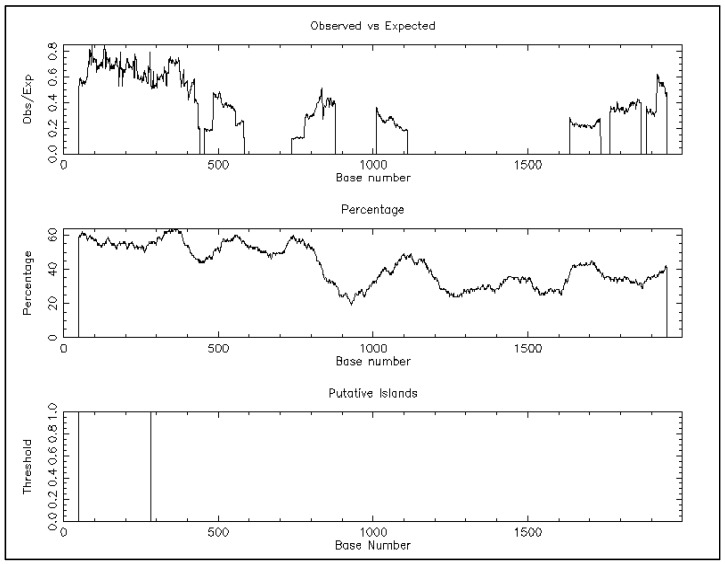
Promoter CpG island analysis of the sheep *MC4R* promoter by the EMBOSS CpG plot (http://www.ebi.ac.uk/Tools/seqstats/emboss_cpgplot/) with a window size of 100 bp and the following set options: observed/expected ratio > 0.60, percent C+ percent G > 50.00, and length > 200. The DNA sequences of the MC4R gene used here contain 2.00 kb before the start codon. Only one CpG island (234 bp) was detected within the sequence −60/−293.

**Figure 3 ijms-20-00240-f003:**
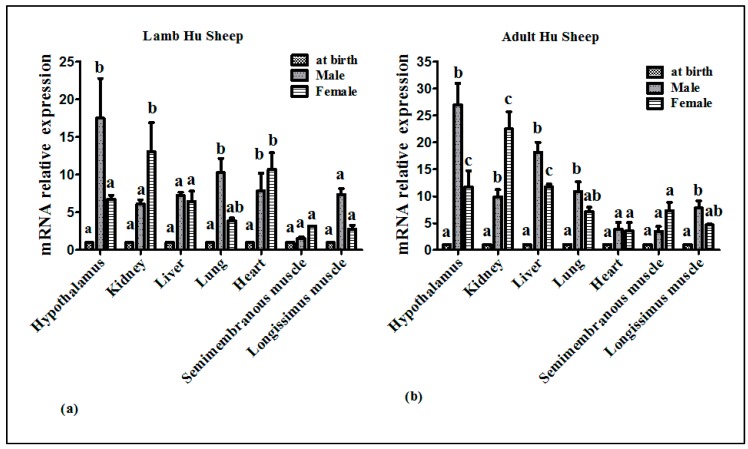
Differential *MC4R* mRNA expression in seven different tissues of Hu sheep (*N*= 42). (**a**) The lamb (one to three months) relative mRNA expression of *MC4R* (*N* = 21). (**b**) The adult (four to six months) relative mRNA expression of *MC4R* (*N* = 21). The mRNA expression of the *MC4R* gene at birth was used as a calibrator. Different superscripts (a, b, and c) are significantly different at *p* < 0.05.

**Figure 4 ijms-20-00240-f004:**
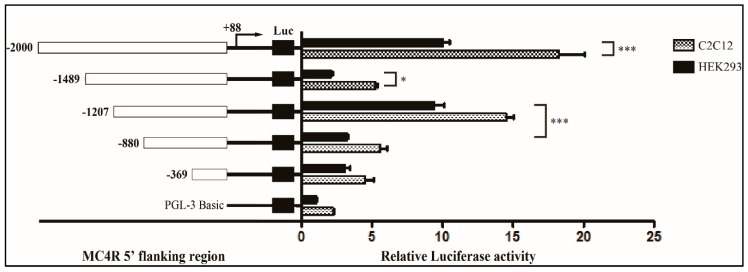
Luciferase activities of the sheep *MC4R* promoter construct in HEK293 and C2C12 cell lines. A series of plasmids containing 5′ unidirectional deletions of the promoter region of the *MC4R* gene (pGL3-2000, pGL3-1489, pGL3-1207, pGL3-880, pGL3–369, and pGL3-Basic) fused in frame to luciferase gene were transfected into HEK293 and C2C12 cell lines. After 48 h, the cells were harvested for luciferase assay. The results are expressed as the mean ± standard deviation in arbitrary units based on the firefly luciferase activity normalized against the Renilla luciferase activity for triplicate transfections. The error bars denote the standard deviation. The two-way analysis of variance (ANOVA) test was used to detect significance; *** *p* < 0.01, **p* < 0.05.

**Table 1 ijms-20-00240-t001:** Single nucleotide polymorphisms (SNPs), alleles, and corresponding genotypes detected in the *melanocortin 4 receptor* (*MC4R*) gene of *Ovis aries*. MAR—minor allele frequencies.

Locus	*N*	Genotype Frequency			MAF	χ2(HWE)	*p*-Value
		CC	CT	TT	T		
−1131C>T	206	0.54(112)	0.35(72)	0.11(22)	0.28(116)	3.814	0.051
		GG	GA	AA	A		
−1038G>A	206	0.63(130)	0.30(62)	0.07(14)	0.22(90)	2.896	0.089
		GG	GT	TT	T		
−1036G>T	204	0.59(120)	0.31(63)	0.1(21)	0.26(105)	7.526	0.006
		GG	GA	AA	A		
−1026G>A	205	0.6(122)	0.34(69)	0.07(14)	0.24(97)	0.954	0.329
		GG	GT	TT	T		
−943G>T	206	0.66(136)	0.26(54)	0.08(16)	0.21(86)	8.780	0.003
		GG	GA	AA	A		
−287G>A	206	0.56(115)	0.34(72)	0.09(19)	0.27(110)	2.360	0.124
		GG	GA	AA	A		
−206G>A	206	0.64(131)	0.36(75)	0.0(0)	0.18(75)	10.203	0.001
		CC	CG	GG	G		
−103C>G	206	0.59(121)	0.33(69)	0.08(16)	0.25(101)	1.857	0.173

*N*: Number of sheep involved in the analysis: Hardy–Weinberg equilibrium (HWE), *p* < 0.05 significant.

**Table 2 ijms-20-00240-t002:** Estimated association effects of eight SNPs in the *MC4R* gene promoter on the body measurement traits in Hu sheep (least-squares mean (LSM) ± standard error of means (SEM)).

Locus	N	Genotype	Body Measurement And Fat Deposition Traits							
			BW	HW	HG	BL	RL	RW	BFT	LIP
−1131C>T	206	CC	23.05 ± 0.17	55.18 ± 1.00 ^a^	59.54 ± 1.37 ^a^	50.72 ± 1.10 ^a^	14.07 ± 0.30 ^a^	13.08 ± 0.07 ^a^	0.66 ± 0.01	4.89 ± 0.21
		CT	25.73 ± 0.47	58.26 ± 0.76 ^b^	61.58 ± 1.00 ^ab^	51.97 ± 0.79 ^a b^	15.42 ± 0.18 ^b^	14.53 ± 0.13 ^b^	0.73 ± 0.03	5.57 ± 0.33
		TT	25.91 ± 0.10	58.22 ± 0.99 ^b^	62.36 ± 1.36 ^b^	52.90 ± 1.10 ^b^	15.48 ± 0.24 ^b^	14.44 ± 0.23 ^b^	0.74 ± 0.03	5.60 ± 0.78
*p*			1.31	0.31	0.04	0.04	0.00	0.00	0.16	0.17
−1038G>A	206	GG	23.14 ± 0.16 ^a^	55.84 ± 0.85	59.63 ± 1.15	50.76 ± 0.88	14.18 ± 0.08	14.23 ± 0.19	0.69 ± 0.04	4.85 ± 0.55 ^a^
		GA	26.46 ± 0.59 ^b^	56.14 ± 0.76	60.33 ± 1.04	50.61 ± 0.80	15.61 ± 0.19	14.58 ± 0.17	0.77 ± 0.03	5.74 ± 0.49 ^a b^
		AA	25.44 ± 0.64 ^b^	57.14 ± 1.27	60.42 ± 1.72	50.92 ± 1.32	15.56 ± 0.24	14.61 ± 0.31	0.71 ± 0.05	6.12 ± 0.82 ^b^
*p*			0.02	0.11	0.36	0.49	0.26	0.62	0.07	0.04
−1036G>T	204	GG	25.93 ± 0.15	56.77 ± 0.78	60.55 ± 1.10	50.58 ± 0.81 ^a^	14.11 ± 0.18 ^a^	13.14 ± 0.17 ^a^	0.67 ± 0.03 ^a^	4.88 ± 0.51
		GT	26.45 ± 0.58	59.26 ± 0.86	60.33 ± 1.20	53.98 ± 0.89 ^b^	15.71 ± 0.21 ^b^	14.70 ± 0.19 ^b^	0.76 ± 0.04 ^b^	5.64 ± 0.57
		TT	25.31 ± 0.57	56.71 ± 0.93	60.33 ± 1.30	50.52 ± 0.96 ^a^	15.04 ± 0.22 ^c^	14.16 ± 0.21 ^c^	0.66 ± 0.04 ^a^	5.79 ± 0.61
*p*			0.17	0.23	0.84	0.01	0.00	0.02	0.05	0.19
−1026G>A	205	GG	23.05 ± 0.16 ^a^	57.88 ± 0.90	59.48 ± 1.22	50.67 ± 0.93	14.10 ± 0.21 ^a^	13.12 ± 0.21 ^a^	0.76 ± 0.04	5.89 ± 0.59
		GA	26.08 ± 0.48 ^b^	57.82 ± 0.71	60.18 ± 0.96	51.30 ± 0.74	15.59 ± 0.17 ^b^	14.64 ± 0.16 ^b^	0.75 ± 0.03	5.50 ± 0.47
		AA	26.39 ± 1.45 ^b^	57.22 ± 1.11	60.93 ± 1.50	51.79 ± 1.15	15.64 ± 0.26 ^b^	14.60 ± 0.25 ^b^	0.76 ± 0.05	5.27 ± 0.73
*p*			0.00	0.99	0.52	0.98	0.00	0.00	0.07	0.19
−943G>T	206	GG	23.16 ± 0.16 ^a^	55.91 ± 0.84 ^a^	59.58 ± 1.15 ^a^	50.55 ± 0.88 ^a^	14.19 ± 0.21 ^a^	13.24 ± 0.19 ^a^	0.77 ± 0.04	4.95 ± 0.54 ^a^
		GT	26.85 ± 0.64 ^b^	59.12 ± 0.79 ^b^	63.03 ± 1.10 ^b^	53.90 ± 0.83 ^b^	15.67 ± 0.19 ^b^	14.64 ± 0.18 ^b^	0.76 ± 0.04	5.43 ± 0.52 ^a b^
		TT	25.34 ± 0.78 ^b^	58.12 ± 1.16 ^b^	63.62 ± 1.60 ^b^	53.18 ± 1.20 ^b^	15.84 ± 0.29 ^b^	14.89 ± 0.27 ^b^	0.72 ± 0.05	6.57 ± 0.75 ^b^
*p*			0.00	0.03	0.03	0.00	0.00	0.00	0.84	0.05
−287G>A	206	GG	25.02 ± 0.18	57.92 ± 1.10 ^a^	59.71 ± 1.39	52.72 ± 1.10	14.11 ± 0.25 ^a^	13.14 ± 0.24 ^a^	0.71 ± 0.05	4.92 ± 0.66
		GA	25.70 ± 0.44	57.45 ± 0.72 ^b^	60.29 ± 0.97	52.68 ± 0.74	15.33 ± 0.18 ^b^	14.37 ± 0.17 ^b^	0.72 ± 0.03	5.45 ± 0.46
		AA	25.66 ± 0.12	57.21 ± 1.10 ^a b^	60.19 ± 1.14	53.21 ± 1.14	15.90 ± 0.27 ^c^	14.93 ± 0.25 ^c^	0.73 ± 0.05	5.96 ± 0.71
*p*			0.21	0.23	0.96	0.16	0.01	0.00	0.66	0.17
−206G>A	206	GG	23.38 ± 0.23	55.86 ± 0.34	59.34 ± 0.38	50.89 ± 0.35	14.30 ± 0.09	13.33 ± 0.09	0.68 ± 0.01	5.01 ± 0.22
		GA	25.89 ± 0.45	58.49 ± 0.51	62.48 ± 0.81	52.95 ± 0.55	15.41 ± 0.14	14.43 ± 0.13	0.73 ± 0.02	5.55 ± 0.32
		AA	-	-	-	-	-	-	-	-
*p*			0.01	0.00	0.00	0.00	0.00	0.00	0.03	0.15
−103C>G	206	CC	23.04 ± 0.17 ^a^	55.73 ± 0.88 ^a^	59.53 ± 1.20 ^a^	50.62 ± 0.92 ^a^	14.11 ± 0.22 ^a^	13.14 ± 0.21 ^a^	0.67 ± 0.04	4.80 ± 0.57 ^a^
		CG	25.94 ± 0.48 ^b^	58.69 ± 0.75 ^b^	62.91 ± 1.02 ^b^	53.17 ± 0.78 ^b^	15.48 ± 0.18 ^b^	14.52 ± 0.17 ^b^	0.73 ± 0.03	5.44 ± 0.46 ^a^
		GG	26.65 ± 1.25 ^b^	58.31 ± 1.20 ^b^	61.25 ± 1.69 ^a b^	52.68 ± 1.20 ^a b^	15.84 ± 0.31 ^b^	14.81 ± 0.29 ^b^	0.75 ± 0.05	7.17 ± 0.81 ^b^
*p*			0.05	0.02	0.03	0.03	0.00	0.00	0.65	0.00

BW: body weight (kg); HW: height at withers (cm); HG: heart girth (cm); BL: body length (cm); RL: rump length (cm); RW: rump width (cm); BFT: backfat thickness (cm); LIP%: lipid percentage of wool. Values with different superscripts (^a^,^b^,^c^) within the same row in the same locus denote significant difference (*p* < 0.05).

**Table 3 ijms-20-00240-t003:** Association analysis of the haplotype combination with body measurement and fat deposition traits in Hu sheep.

Diplotype	Frequency	Body Measurement and Fat Deposition Traits (Mean ± SE)		
		BW	BFT	LIP%
H1H1 (CC-GG-GG-GG-GG-GG-GG-CC)	49.5% (102)	22.86 ± 0.16 ^a^	0.67 ± 0.02 ^a^	4.73 ± 0.23 ^a^
H1H2 (CT-GA-GT-GA-GT-GA-GA-CG)	17.47% (36)	26.49 ± 0.78 ^b^	0.76 ± 0.03 ^b^	5.91 ± 0.45 ^ab^
H3H3 (TT-AA-TT-AA-TT-AA-GG-GG)	7.28% (15)	24.53 ± 0.69 ^c^	0.71 ± 0.04 ^ab^	6.24 ± 0.88 ^b^

Note: means with different superscripts (^a^, ^b^, and ^c^) in the same column are significantly different at *p* < 0.05.

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
