# Peer review of "Variation in the Promoter Region of the MC4R Gene Elucidates the Association of Body Measurement Traits in Hu Sheep"

_ijms, 2019, doi:10.3390/ijms20020240_

Round 1
Reviewer 1 Report
The paper reports the assessment of polymorphisms in the promoter region of the gene
Melanocortin 4 receptor (MC4R) in sheep. In Addition, the authors assessed the association of the polymorphisms with performance records, expression of mRNA in diverse tissues of males and females of two different ages. Finaly, the authors investigated the most active region of the promoter to narrow down the potential regulators of MC4R. The paper is mostly well organized, the paper is well structured, but the writing needs to be improved. Also a few aspects must be revised to make it a document worthy of publication.
The sentences that contain the abbreviation “I.e,” must be changed to appropriate grammatical form.
There is a discrepancy between the sentence: “no polymorphisms of the MC4R gene 61 promoter in sheep have been reported” and “Among these, two SNPs (-1038G>A, and -74 943G>T) were novel”. If six had been already identified, how is it that no polymorphism have been reported?
As you write to the reader “(-1131C>T; -1038G>A; -1036G>T; -1026G>A; -73 943G>T; -287G>A; -206G>A; and -103C>G)” on subheading 2.1, indicate what is nucleotide zero.
On Table 1, under p-value “ns” and “**” are unnecessary and just make for a busy colum. I’d suggest that you remove them since you have the p values
Most of the databases/software you used have a paper. Use citation instead of their website.
Table 2 header. What is this? “element/s”
Line 126-129: 25.89+0.45kg and others. Did you want to use plus and minus sign?
Table 3 must be reformatted. It is too hard to read it the way it is now. Also, there seems to be some problems with plus and minus sign.
Table 4. Why do you have four decimals? It also needs to be better formatted.
Figure 3. I do not understand why you added bisulfite primers on that figure.
I have a serious problem with you inferring a sequence conserved by using only one species for comparative purposes. Most work would use many more mammalian sequences to assess conservation.
In the methods you stated a general linear model to analyze BMTs. Can you do regular GLM on percentage without transformation, for instance on Lip%?
Line 212: “from day old”. What is day old? Did you mean one day old? Please fix this.
Under subheading 4.4, you did not state the statistical tests used for comparing mRNA relative levels. Also on that methods section, what sample was used as a baseline, which should show expression around 1 on charts on Figure 5.
Author Response
Dear reviewer:
Please find the comments in the attachment
Thanks.

Reviewer 2 Report
This submitted study by Girmay and colleagues describes the identification of polymorphic variation in the promoter region of melanocortin 4 receptor (MC4R) gene in Hu sheep, which is not reported early. With the analysis of DNA samples from 206 Hu sheep, authors have discovered 8 regulatory SNPs and 11 haplotypes, of which 2 SNPs were novel. All the SNPs were significantly associated with linear body measurement traits in sheep. Among the identified haplotypes, H1H2 had heavier body weight, and thus this haplotype may serve as a genetic marker to optimize breeding programs for body measurement traits in Hu sheep.
The manuscript has been well-written with the originally obtained data. However, authors need to revise several parts of the manuscript, and prepare additional data to strengthen the manuscript.
Specific points.
1. The manuscript is currently looks like a bioinformatics paper, because most of the data were computer analyzed, also the figures are computer generated. I suggest the authors to reduce many information which has less scientific value in this study (including “data not shown” information).
2. The articles by Song et al., 2012 and Zuo et al., 2014 (Ref. no 15 and 16), which also reported polymorphisms in the MC4R gene in sheep, are not well-discussed in this paper. Although those articles are not related to the promoter region, the polymorphisms and associated traits reported in those studies should be well-discussed by comparing the present investigation. Similar comment to the below articles that are not cited in the submitted manuscript.
Sanz et al., 2015; https://doi.org/10.1016/j.smallrumres.2014.10.010
Wang et al., 2015; https://doi.org/10.1016/j.smallrumres.2015.02.007
3. Results section 2.6 should be rearranged and strengthen with more data. The MC4R expression in 7 tissues from male and female lamb should be given in “a”, and the MC4R expression in 7 tissues from male and female adult should be given in “b” (please specify correct age in “a” and “b”).
4. I strongly recommend the authors to perform in situ hybridization or immunochemistry to examine the expression pattern of MC4R in the brain parts (including hypothalamus), kidney, liver, heart, lung, and muscle of adult sheep (male or female). Because such detailed expression pattern is not available in sheep to my understanding (except for Neuroscience. 2001;105(4):931-40), and your paper will receive many attractions once it published.
5. The authors repeatedly mentioned that they have analyzed 2088 kb promoter region upstream of ATG. I think, the analysis region is 2000 kb upstream and 88 kb downstream of ATG in the CDS? Please revise correctly throughout the manuscript.
6. Lines 378-380, The sampling procedure is not clear. Seven tissues from day-old and 6-months separately? or day-old to 6-months (0, 1, 2, 3, 4, 5, 6-months)? What is n=42?
7. Section 4.3. Please write clearly the statistical methods used for each analysis. For instances, the ANOVA procedure is not written here, and it is not known which statistics used for qPCR.
Author Response

(The authors gave the same response as above.)

Round 2
Reviewer 1 Report
Table 1 still needs revision. For instance, I should not have to, but I am assuming that the number between parenthesis is number of individuals on the first SNP, 112 + 72 + 12 adds up to 196, not 204.
I do not understand why the authors used website address instead of tool name on item 4.5
On their rebuttal the authors presented a histogram with the distribution of the observations for lipid percentage. That only made me even more skeptical about it. I look at the distribution with average 0.05 and standard deviation 0.025. Then I look at table 2 and see that the LSM (which should not be too much different from the arithmetic mean), and I read numbers between 4 and 8.
Author Response
1.Table 1 still needs revision. For instance, I should not have to, but I am assuming that the number between parenthesis is number of individuals on the first SNP, 112 + 72 + 12 adds up to 196, not 204.
Reply: Thank you for constructive comment and we apologize for the typing error. The typing error in table 1 is corrected in the revised manuscript (Table 1, page 3 of 17).
.
2. I do not understand why the authors used website address instead of tool name on item 4.5
Reply: We have accepted your valuable comment. In accordance with the comment we have included the tool names beside of the website address for ease of access for readers in the revised manuscript (line 425 to 431; Page 12 of 17).
3.On their rebuttal the authors presented a histogram with the distribution of the observations for lipid percentage. That only made me even more skeptical about it. I look at the distribution with average 0.05 and standard deviation 0.025. Then I look at table 2 and see that the LSM (which should not be too much different from the arithmetic mean), and I read numbers between 4 and 8.
Replay:
We are grateful for your critical insight and we apology for the discrepancy, possibly that makes you feel skeptical. However, this is not the mistake in our data rather in the table the data were put in the percentage whereas; in the histogram the data were expressed in the ratio. That is the reason that the mean of the data presented in the table is different from the mean of the data presented in the histogram, nevertheless, if we multiply the mean of the histogram (0.05) which is presented in the ratio by 100, we will get the 5% which is similar to the mean of the data presented in the table. For more clarity, we have attached the raw data that we used to construct the histogram in the ratio form and the percentage data that we presented in the table 2 in the percentage form. Hopefully, this will solve the confusion about the difference in the mean of the histogram and table 2 in the manuscript (Table 2; page 5 of 17).

Reviewer 2 Report
The files shows that the authors have corrected most of my comments.
Author Response
Thank you for your valuable comments.